# Is Gender an Antecedent to Workplace Stressors? A Systematic Review and an Empirical Study Using a Person-Centred Approach

**DOI:** 10.3390/ijerph20085541

**Published:** 2023-04-17

**Authors:** Roberta Fida, David Watson, Valerio Ghezzi, Claudio Barbaranelli, Matteo Ronchetti, Cristina Di Tecco

**Affiliations:** 1Norwich Business School, University of East Anglia, Norwich NR4 7TJ, UK; 2Department of Psychology, Sapienza University of Rome, 00185 Rome, Italy; 3Department of Occupational and Environmental Medicine, Epidemiology and Hygiene, Italian Workers’ Compensation Authority (INAIL), Monte Porzio Catone, 00078 Rome, Italy

**Keywords:** gender, stressors, workplace stress, systematic review, latent profile analysis

## Abstract

Objective: Work is a key domain of life in which gender inequality can manifest, yet gender is rarely the explicit focus of research seeking to understand exposure to stressors. We investigated this research gap in two studies. Methods: Study 1 was a systematic review of the relationship between gender and key stressors (e.g., high demands, poor support, lack of clarity and control). From a total of 13,376,130 papers met our inclusion criteria. Study 2 was a cross-sectional study that included 11,289 employees nested within 71 public organisations (50.6% men). Through a latent profile analysis, we investigated the profiles of stressors separately from men and women. Results: The systematic review revealed that, for all stressors, a significant proportion of studies found no significant gender differences, and the review found mixed evidence of greater exposure for both men and women. The results of Study 2 revealed that both genders could be optimally represented by three psychosocial risk profiles reflecting medium, low and high stressors. The results also showed that while the shape of profiles was similar for both genders, men had a higher probability than women of being in the *virtuous* (i.e., low stressors) profile, and the opposite pattern emerged for the *average* profile (i.e., medium levels of stressors). Men and women displayed the same likelihood of being classified in the *at-risk* profile (i.e., high levels of stressors). Conclusion: Gender differences in exposure to stressors are inconsistent. Although the literature on gender role theory and the gendering of work suggests different exposures to stressors in men and women, we find little empirical support for this.

## 1. Introduction

Gender equality remains a key issue for society [1]. Work is a key domain of life in which gender inequality can be manifested, for example, in terms of the pay gap [2,3] and career opportunities [4,5]. However, gender is rarely the explicit focus of research in seeking to understand exposure to workplace stressors, with gender generally limited to being regarded as an important co-variate alongside other demographic variables.

Literature exploring the relationship between gender and exposure to workplace stressors is equivocal [6]. The lack of a clear relationship between gender and exposure to workplace stressors suggests that the intersection of gender and work context or job design is likely to be important in determining exposure. We aim to investigate this important gap by conducting a systematic review (Study 1) and empirical study (Study 2) on how gender influences exposure to a more or less stressful work environment using a large data set of Italian public administration workers—a context where we expect less difference in work role and environment according to gender.

### 1.1. Stress and Gender

There is substantial literature linking gender to stress and burnout [7,8]. Most research has examined gender differences in stress reactions rather than exposure to stressors [9,10]. For example, one meta-analysis found that women experienced higher levels of emotional exhaustion than men, while men were somewhat more depersonalised than women [8]. Similarly, a study on the effect of night shifts on diabetes found a stronger impact on women [10]. This literature suggests that women and men differ from each other depending on the specific measures of stress but tells us little about exposure to stressors. Moreover, findings are inconsistent; another meta-analysis [11] showed that high psychological demands were positively related to common mental disorders at a moderate risk level, and this was higher for men than women, whereas a recent meta-analysis identified an increased likelihood of burnout development in nurses who are men [12]. Overall, the research shows inconsistent results for differences in perceived or experienced work stress between men and women, and longitudinal data similarly reflect a lack of difference [13].

While the findings are inconsistent, different reactions indicate that gender influences how stress is perceived. Consequently, we expect the perceptions of stressors to vary. Some research suggests that stressors can be experienced differently by men and women because of differences not just in perception but also in how they are exposed to stressors, as well as differences in coping. For example, women are more likely to work part-time and occupy roles that are lower in the organisational hierarchy [14]. The literature also suggests that women and men are exposed to different working conditions [15] and are subjected to different types of demands, even when working in the same industry or profession.

Gender differences in exposure and perception of stressors can be summarised as what Marchand et al. [16] referred to as the *vulnerability* and *exposure* hypotheses. The former attributes greater vulnerability to stress based on biological predisposition and socialisation into gender-specific roles and identities that shape expectations and interpretations of work conditions. The latter posits that exposure to psychosocial stressors is shaped by the way the workplace is gendered, determining the kinds of roles and occupations men and women do and how they may be treated in those roles according to gender [2,17]. While both hypotheses offer a plausible explanation for gender differences in the experience of work stress, the empirical data do not provide a consensus [16]. Despite clear evidence of segregation in the labour market, the negative consequences cannot be inferred [2].

### 1.2. Theorising Exposure to Work Stressors According to Gender

A growing body of literature recognises psychosocial stressors as complex phenomena that result from the interaction between job content, work organisation and management, and other environmental and organisational factors with employees’ competencies and coping ability [18]. Consistently, the most supported theories in the work and organisational literature consider workplace stress to be a constellation of working conditions. Gender theory suggests that men and women adopt, learn and perceive roles according to gender and that this is influenced by social, psychological and biological factors. Consequently, we expect that their perceptions, coping strategies and behaviours in the workplace will be influenced by gender expectations and lead to gender differences based on work roles and conditions. Accordingly, this study set out to investigate *whether exposure to psychosocial risk factors differs for men and women.*

With the present study, we thus contribute to the literature on gender and stress in several ways. First, we report on a systematic review (Study 1) to explicitly explore the relationship between gender and psychosocial risk factors at work to understand and summarise gender differences in stress exposure for key stressors. Our systematic review also identifies important research gaps in the stress literature. Second, we go beyond previous research on gender-specific stressor exposure in investigating the existence of profiles for women and men based on *constellations* of stressor exposure variables rather than isolated stressors. We use a person-centred approach to identify naturally occurring profiles by analysing a large data set of Italian public administration workers (Study 2). Studies that investigate the relationship between gender and stressor exposure usually examine single variables that may mask the heterogeneous profiles of potentially differentiated individuals. Conversely, we examine gender-specific multidimensional profiles to investigate whether belonging to one of these gender sub-groups is associated with lower or higher levels of stressor exposure. The advantages of such a person-centred approach are not only methodological [19] but also conceptual, as they are more in line with the theoretical understanding of gender roles described above.

## 2. Study 1: Systematic Review

The aim of this systematic review is to better understand gender differences in exposure to psychosocial work stressors. While most recent systematic reviews have focused on gender differences in mental health experience [12,16], almost no work has been done in relation to exposure to psychosocial stressors at work. To the best of our knowledge, the only recent systematic review in this area has been done in relation to work–family conflict [20]. Therefore, we do not include studies assessing work–family conflict in the final review, and we focus specifically on key psychosocial risks identified by the literature [21,22]: job demands, job control, supervisor and colleagues support, relationships at work, role stressors, and change.

### 2.1. Study 1 Methods

#### 2.1.1. Search Terms and Inclusion/Exclusion Criteria

The acronym PICOS (population, intervention, comparators, outcome, study design) is typically used as a way of understanding different elements of a research question for a systematic review and to inform the development of search terms and inclusion/exclusion criteria. However, not all systematic reviews focus on intervention studies for which the PICOS approach has been designed. Following Booth’s recommendation to take the best approach suited to the individual review [23], we specified the focus of the review and developed search terms according to population, *phenomenon of interest* (instead of intervention), relevant comparators, outcomes and study design [1] (Please see study protocol for detailed methods, search term strategy and details of inclusion/exclusion criteria, which is reported in accordance with the Preferred Reporting Items for Systematic Review and Meta-Analysis Protocols (PRISMA-P) Statement [24]. Short version of protocol registered (CRD 42018110892) and available at https://www.crd.york.ac.uk/prospero/display_record.php?ID=CRD42018110892 (full version available online at https://www.researchgate.net/publication/369386077_The_relationship_between_gender_and_risk_factors_for_psychosocial_stress_in_the_workplace_A_systematic_Review_PRISMA-P_1a).

For the population, we focused on the general working population. Given the context of Study 2 research (Italy), we included studies in a similar developed economic context (e.g., EU-15 countries, USA, UK, Australia, Korea, Japan, etc.). The phenomenon of interest was related to the prevalence of or exposure to different psychosocial stressors in the workplace. Gender was the comparator; hence, only studies sampling both men and women were included. The outcomes considered in the systematic review were the different stressors, understood as psychosocial factors contributing to stress in the workplace. We focused on job demands, job control, supervisor and colleague support, relationships at work, role stressors, and support for change. We did not include papers on work–family conflict given the recent published meta-analysis on this topic, which in general found greater evidence for similarity than difference in exposure to work–family conflict [20]. Initially, studies using quantitative and qualitative methods were included in the review, but we only present findings from the quantitative studies in this paper that informed our quantitative analysis. We included only empirical research published in a peer-reviewed journal that met the inclusion criteria specified above. We included studies not published in English in our searches, but only those that the review team had language skills to translate were included in the final papers because of time and resource constraints.

#### 2.1.2. Study Selection

Studies that met the inclusion criteria were sifted according to the following process. Titles and abstracts were reviewed by at least two members of the team and assessed as to whether they met inclusion criteria. In cases of disagreement, a third reviewer made a decision and if any doubt remained, the paper was retained for full paper screening. At the full paper stage, one reviewer assessed the papers’ relevance and recorded the decision and justification for inclusion/exclusion. A second reviewer double checked 10% of these decisions and two members of the review team, who extracted data, double checked all included studies for relevance. Given the very high number of studies identified, we restricted the review to studies published within the last 5 years of searches being completed (2012–2017). The following flow diagram (Figure 1) shows the progression of the review and the eventual number of studies included.

#### 2.1.3. Data Extraction and Evidence Quality

Data extraction sheets were piloted by each member of the review team by reading through and extracting papers from two papers independently. The findings and any differences were discussed, and the data extraction sheet was modified. Each paper was extracted by one member of the review team. Of each reviewers’ papers, 10% were read by another member of the team to ensure consistency of interpretation.

A quality statement was included as part of the data extraction sheet for each full paper. This statement was informed by best practice quality checklists [25]. The overall quality grading for the review findings was informed by the GRADE approach specified in the Cochrane Centres handbook [26]. Accordingly, quality was graded according to four categories: high, moderate, low and very low.

### 2.2. Study 1 Results

From a total of 130 papers screened, 66 met the inclusion criteria (See Appendix A
Table A1 and Table A2). Because three papers, 87, 88 and 89, used the same data, we counted them only once when reporting results. Many of the papers reported more than one stressor. Overall, 44 papers (67%) investigated job *demands*, either psychological, physical or emotional, as well as rewards; 36 papers (54%) investigated the role of autonomy and *control*; 33 papers (50%) investigated the role of *support* from the supervisor, colleagues and/or others; 30 papers (45%) investigated the quality of *relationships* at work in terms of positive collaborations, as well as workplace bullying, aggression, violence and harassment from supervisors, colleagues and/or others; 10 papers (15%) investigated role stressors in terms of both *role* conflict and role clarity. Overall, among the 66 studies included, 21 of them (32%) used large samples (>4000 employees), 16 of them (24%) used medium to large samples (1000–4000 employees), 26 studies (39%) used smaller samples (200–1000 employees), and 3 studies (4%) used samples that included less than 200 employees. The quality of the studies included is quite high, with 52 (79%) rated high quality, 12 (18%) rated moderate quality and only two (3%) considered low quality. Appendix A
Table A1 presents a summary broken down by the different themes of literature that emerged from the studies included in this systematic review.

#### 2.2.1. Job Demands and Rewards

We identified a total of 44 studies (65%) investigating different forms of job demands (either psychological, physical or emotional) and/or rewards (in terms of either effort-reward balance or developmental opportunities) in relation to gender. The results were inconsistent, but the weight of evidence suggested no gender differences between men and women in relation to job demands and rewards but more exposure to emotional demands for women. Out of the 42 studies measuring demands or workload, 23 (55%) of them showed no significant gender difference in at least one indicator, 13 (31%) reported that men perceived higher demands and 10 studies (24%) reported the opposite with women perceiving higher demands. Three studies specifically explored physical demands, and two of them (67%) showed no gender differences, and one study showed higher levels for men (68). In relation to emotional demands, out of the seven studies identified, all of them showed higher exposure for women. Finally, in relation to rewards, out of the 14 studies investigating it, 10 (71%) reported no significant gender differences, 3 (21%) reported higher rewards for men, and only 1 (7%) reported higher levels for women.

#### 2.2.2. Relationships at Work: From Positive Collaboration to Harassment and Violence

We identified 30 studies presenting the results of the different forms of relationships in relation to gender. The studies identified explored different types of relationships, from positive collaboration to more relationships, such as interpersonal aggression and violence. The results again showed an inconsistent picture in relation to the role of gender. Indeed, a significant proportion of studies (23 studies, 77%) reported no significant gender difference, especially in relation to positive indicators, such as quality of relationships and collaboration. Twelve studies (40%) reported women to be more exposed to some forms of bullying or more severe forms of negative interpersonal behaviour; however, six studies (20%) showed the opposite (higher for men). When looking at the more severe forms of aggression, among the three studies (10%) examining physical aggression, none reported gender differences. Sexual harassment was explored in five studies (17%), and three reported higher exposure for women, one the opposite (109) and one no gender difference.

#### 2.2.3. Role Stressors

We identified 10 studies presenting the results of the different forms of role stressors in relation to gender. The studies identified explored different types of role stressors, such as role overload, role conflict and role ambiguity. Most of the studies (70%) showed no gender differences, at least in some indicators. However, there was also evidence of higher exposure for women, but one study also reported the opposite (higher for men).

#### 2.2.4. Support

We identified 33 studies evaluating gender differences in the exposure to different types of support. Overall, the findings presented a mixed picture, but the weight of evidence suggested no gender differences. Among the 13 studies examining a broad dimension of social support at work, five of them (38%) reported no gender differences; however, five studies (38%) found that women perceived higher support, and three studies (23%) found the opposite with men perceiving higher social support. A total of 15 studies examined gender differences in supervisor support. Among them, 11 (73%) reported no gender differences; however, four studies (17%) found that men perceived higher supervisor support, and no studies reported higher levels for women. A total of 15 quantitative studies examined the quality of support from colleagues. Among them, 11 studies (73%) reported no gender differences; however, three studies (27%) found that men perceived higher co-worker support, and one study (0.9%) reported that women perceived higher support. Finally, seven studies in this category tended to investigate support outside work in general or family/home support. Overall, the analysis of the results showed that four studies (57%) reported no gender differences, two studies (29%) reported family support to be higher for women, and one paper (14%) reported the opposite (men with higher support from family).

#### 2.2.5. Control

We identified a total of 36 studies measuring some forms of job control and autonomy and their gender differences. Overall, the studies presented a mixed picture. Seventeen studies (47%) found no gender differences, but another fifteen (42%) found that men were exposed to more autonomy and decision latitude. Three studies (8%) found that women experienced higher control.

### 2.3. Study 1: Discussion

We presented brief summaries of our findings in relation to each stressor because the clear common theme across the evidence gathered by the review was a lack of consistent evidence of how gender might influence exposure to key workplace stressors. The review found no significant gender differences for any stressors. While there was evidence for greater exposure to stressors for women in relation to some stressors, such as control, this evidence was at best weak because of the inconsistency of findings. In this regard, we reached a similar conclusion to Shockley et al. in relation to work–family conflict [20]. Popular beliefs about likely exposure to stressors seemed to be at odds with research findings, which were highly inconsistent. However, we could not conclude that gender was not a significant factor in determining exposure since individual studies did show significant differences. It is possible that synthesising findings from multiple studies obscures the role of gender.

The studies identified in the review came from a wide range of contexts; some had very large samples including multiple work sectors. According to origin theories of gender difference, we would expect gender to be important in influencing individuals’ experience of work, and this may be specific to sector. In this sense, certain sectors may be considered ‘gendered’, whereby the sector tends to be dominated by one gender over another. For example, a study included in the review examined exposure to stress in correctional officers, a sector traditionally dominated by men but becoming more gender diverse [27]. While this study only found significant gender differences in role stressors, more generally, it can be argued that the gendered nature of sectors is significant in determining how men and women are exposed to stressors and perceive them in the workplace. Sector-specific effects may be obscured in the synthesis of findings and through the inclusion of large-scale studies including a range of sectors in their sample. It could also be argued that the gendered nature of work is not limited to particular sectors but is more general. A more detailed examination of a sector that is relatively gender balanced in composition and gender neutral in the experience of work would support a rigorous testing of this proposition.

In her seminal article, Acker [28] argued that work and organisations are gendered in a more general sense and that the organisation of work is premised on the division of men as breadwinners and paid workers and women as unpaid caregivers (see also [29]). Therefore, paid work is geared toward masculine norms because of the historic absence of women from the labour force and is therefore more likely to be experienced positively by men and negatively by women who must conform to the ideal of a worker who is a man. If this is the case, then we would expect gender differences in exposure to stressors, regardless of sector. However, a review-level study makes it difficult to test this proposition, particularly since there is inconsistency in the review findings rather than reliably no significant differences between stressor exposure according to gender. Gascoigne et al. argued that the gendered construction of work is crucial in determining exposure to stress in ‘extreme’ jobs, professional and managerial work involving long hours and high demands [29]. In Study 2, we further explored the role of gender in determining exposure to stressors using a gender-balanced sample of public administration workers.

## 3. Study 2: Empirical Study in Public Administrations

In Study 2, consistent with the OECD [30], we examined a sample of public administration (PA) workers where gender composition was evenly split, and we expected not to find gender segregation in role. Much public service work can be highly gender segregated [31], such as police work (dominated by men) in comparison to elementary education (dominated by women), but our sample included PA workers rather than the public sector in general. This sample excluded sectors, such as education, the military, and health and social care, which are gender dominated by men or women; instead, we analysed the experiences of PA workers working in local government and the civil service in Italy. We focused on a large sample of employees (i.e., 11,289 employees nested within 71 organisations), with a specific focus on the organisations from public administration and civil service. Grouping organisations within a single category (PA) allowed us to consider a more homogeneous context in which the proportion of women and men was more balanced.

However, even though PA jobs are not segregated according to explicit gender role expectations, gender may influence roles. For example, men may be considered better suited to cognitive work and women emotionally demanding work and consequently exposed to different stressors [31]. This does not, of course, rule out differences in exposure to stressors, as the gendered division of labour is only one element of what Connell refers to as the gender regimes that determine gender relations in organisations and influence experiences of stress [32]. Connell also identifies gendering of relations of power, emotions and human relations, and culture and symbolism as making up gender regimes in the workplace that might shape the experience of work differently for men and women, thereby exposing them to different types and levels of stressors. Moreover, even where gendered division of labour is not explicit at the sector or occupational level, it may take place at a more micro level within PA worksites or the assignment of tasks [32].

In studying a more occupationally homogenous and gender-balanced workforce, we explored the way in which gender influences exposure to stressors, when it is regarded as more fundamental to organisations [28]. In taking a person-centred approach using latent profile analysis (LPA), we overcame the weaknesses of studies that examine gender differences in single variables, which could have obscured the varied experiences of individuals and the way in which they configured intra-individually. LPA allowed us to group men and women into different profiles to understand whether membership in gender sub-groups was associated with lower or higher exposure to stressors, as well as how exposures to different psychosocial risks were equally (or differently) intertwined within (and across) genders. This approach was in line with our theoretical understanding of the way gender influences stressor exposure. In this sense, although it is difficult to anticipate a specific typology of psychosocial risk exposure for men and women sub-groups, we expect a similar final LPA solution for both genders. Moreover, consistent with our theorising concerning gender-based stress exposure, we expected a different proportion of men and women across similar profiles.

### 3.1. Study 2 Methods

#### 3.1.1. Participants

For the purpose of this study, we used secondary data on Italian employees working in the PA sector collected by the Italian Workers Compensation Authority (INAIL). To support organisations’ legal duty to assess and manage work-related risks, INAIL has developed a methodology that can be freely used by all organisations [33]. This includes the administration of a questionnaire to assess employees’ perceptions of psychosocial risks. Before answering the questionnaire, employees signed an informed consent form to allow INAIL to use the data for research purposes. Individual participation was completely voluntary and unrewarded.

The sample of this study included 11,289 employees nested within 71 organisations (average number of employees per organisation = 161.26, SD = 441.70). The sample gender distribution (50.6% men, 49.4% women) was substantially in line with the most recent Italian census in the field [34]. Almost all of the sample included employees with an Italian nationality (99.8%). Almost half of the sample (49.3%) reported an age of 51 years or above with a slightly higher proportion of women in the 31–50 years category and of men in the 51 years or above category (χ^2^_(*df* = 2)_ = 103.71, *p* < 0.001). Finally, most of the sample reported a permanent employment status (94.3%), with a higher prevalence of women in the contingent employment status (χ^2^_(*df* = 1)_ = 15.67, *p* < 0.001).

#### 3.1.2. Measures

*Psychosocial Risk Factors* were measured by the Italian version of the Management Standards Indicator Tool [35,36]. The tool comprises 35 items originally developed to cover seven psychosocial risk factors: (1) *Role Clarity* measures employees’ understanding of their own role within the organisational context with five items; (2) *Control* measures employees’ degrees of freedom in managing their work activities with six items; (3) *Demands* measures employees’ evaluations of their workload and job pressure with eight items; (4) *Poor Relationships* measures employees’ perceptions of interpersonal conflict and bullying at work with four items; (5) *Peer Support* measures employees’ perceptions of support provided by their colleagues at work with four items; (6) *Management Support* measures employees’ perceptions of support provided by the employer and managers at work with five items; (7) *Change* measures support provided by the employer and managers to the employees regarding how change processes are handled and implemented with three items. Employees rated each item on a five-point Likert-type agreement (from 1 = strongly disagree to 5 = strongly agree) or frequency scale (from 1 = never to 5 = always) depending on the item text, and they were asked to refer to the 6 months prior to questionnaire administration. Items of *Demands* and *Poor Relationships* dimensions were coded so that higher scores reflected higher psychosocial risks, and the opposite was for all other dimensions (i.e., higher scores reflected lower psychosocial risks).

#### 3.1.3. Analytic Approach

Gender differences in the psychosocial risk factor profiles were examined using Latent Profile Analysis (LPA) [37]. For each gender group, we compared the solutions from one to eight profiles. The best fitting solution was identified by taking into account: (1) the Akaike’s information criterion (AIC); (2) the Bayesian Information Criterion (BIC); (3) the Sample Size-Adjusted Bayesian Information Criterion (SABIC); (4) the Vuong–Lo–Mendell–Rubin adjusted likelihood ratio test (VLMR); and (5) the bootstrapped likelihood ratio test (BLRT). The best solution should show low values of AIC and BIC, significant (*p* values < 0.01) VLMR and BLRT associated with a specific LPA, entropy coefficient lower than 70, and with each cluster including at least 10% of the sample.

Once the best LPA solution was established for both genders, profiles’ shapes were compared graphically with the aim of identifying similar profiles between groups. A two-way MANOVA (with profiles’ membership and gender as factors and the psychosocial dimension factor scores as dependent variables) was then carried out to detect gender differences in clustering frequencies and factor scores (as well as profile * gender interactions).

Before conducting the LPA, we followed Morin and colleagues’ approach [38] and compared three alternative factorial structures of the questionnaire separately for men and women: (1) Confirmatory Factor Analytic (CFA) model with oblique factors; (2) Exploratory Factor Structural Equation Model (ESEM) with oblique factors; (3) Fully Symmetrical Bifactor Model (one general factor defined for the entire set of items and seven specific factors, with general and specific factors mutually orthogonal). All models were analysed with the M*plus* 8.4 software [39] with robust maximum likelihood (MLR) estimators and appropriately correcting the parameters’ standard errors for the nested nature of the data [40]. The models were compared considering the AIC and the Expected Cross-Validation Index (ECVI). Lower values indicated better-fitting models. Gender measurement invariance analysis was also carried out [41]. We calculated the factor scores of the most restrictive and tenable measurement invariance model and used them in the LPA and subsequent analyses.

### 3.2. Study 2 Results

#### 3.2.1. Preliminary Analysis

The comparison of the different factorial models attested that the ESEM 7-factor model showed the best fit (lower AIC and ECVI) in both gender groups (men: MLRχ^2^_(*df* = 525)_ = 5212.143; RMSEA = 0.040 (90%C.I. 0.039–0.041), CFI= 0.900 SRMR = 0.046; women: MLRχ^2^_(*df* = 525)_ = 6246.469; RMSEA = 0.044 (90%C.I. 0.034–0.037), CFI= 0.88 SRMR = 0.053) (see Appendix A). However, the results showed a very high correlation between management support and support for change in both gender groups (>0.70). Since collinear indicators might negatively affect the quality of the LPA solutions, we tested an alternative ESEM model positing 6 oblique factors, in line with past evaluation of the MS-IT factorial structure conducted in the Italian context [35]. This model reached a satisfactory fit in both subsamples (Men: MLRχ^2^_(*df* = 400)_ = 3331.934, *p* < 0.001; RMSEA = 0.036 (90% Confidence Interval: 0.034–0.038); CFI = 0.940; SRMR = 0.018; Women: MLRχ^2^_(*df* = 400)_ = 3742.449, *p* < 0.001; RMSEA = 0.039 (90% Confidence Interval: 0.038–0.040); CFI = 0.931; SRMR = 0.022). Thus, the oblique 6-factor ESEM model was retained for both gender groups.

The gender measurement invariance showed that the full strict measurement invariance model was fully tenable (see Appendix A), with no gender differences in the factor loadings, residual variances, and intercepts. Thus, the LPA was conducted using the factor scores derived from the strict 6-factor ESEM model. Table 1 reports the correlations among the factors, as well as the reliability coefficients (i.e., the composite reliability ω and the maximal reliability *H* coefficients [42]. All correlations were significant in both the men’s and women’s groups. The reliability coefficients were satisfactory in both groups (>0.70). Finally, we tested for gender differences in these dimensions [*F*_(6;11,278)_ = 4.302, *p* < 0.001; Wilk’s Λ = 0.998, partial η^2^ = 0.002], and the results showed a very low multivariate effect size (albeit significant, due to the large sample size) and a univariate effect size lower than 0.01%, suggesting no substantial differences in any psychosocial risk dimension between men and women.

#### 3.2.2. Gender Profiles in Psychosocial Risk Factors: Results of Latent Profile Analysis

Table 2 displays fit indices of the LPA solutions separately for gender. The analysis of the information criteria [43,44] in both gender subgroups (Figure 2) indicated that the 3-profile LPA was the best solution (Figure 3).

Profile 2 (38% of the men subgroup, 43.2% of the women subgroup, 40.5% of the total sample) was interpreted as *normative*, with medium levels in all dimensions. Profile 3 (52% of the men subgroup, 46.7% of the women subgroup, and 49.4% of the total sample) was interpreted as the *virtuous* profile, showing the highest perceived levels of role clarity, control, peer and management support, and the lowest perceived levels of demands and poor relationships. Profile 1 (10% of the men subgroup, 10.1% of the women subgroup, 10.1% of the total sample) mirrored the previous group, and it was interpreted as the *at-risk* profile. Employees in this profile showed the lowest perceived levels of role clarity, control, peer and management support, and the highest perceived levels of demands and poor relationships.

Table 3 shows the results concerning the association between profile membership and gender. The difference in gender proportions associated with Profile 1 was not statistically significant, suggesting that men and women had the same probability of being clustered in the at-risk profile. Finally, women showed a statistically significant higher probability than men of being clustered in the average profile (Profile 2), while the opposite gender pattern was observed for the virtuous profile (Profile 3).

Finally, the factorial two-way MANOVA analysis showed statistically significant multivariate effects for profile membership [*F*_(12;22,556)_ = 3885.834, *p* < 0.001; Wilk’s Λ = 0.189, partial η^2^ = 0.565], gender [*F*_(6;11,278)_ = 17.052, *p* < 0.001; Wilk’s Λ = 0.991, partial η^2^ = 0.009] and their interaction [*F*_(12;22,556)_ = 15.133, *p* < 0.001; Wilk’s Λ = 0.984, partial η^2^ = 0.008]. Table 4 reports all principal effects of this analysis. The analysis of univariate simple effects showed some gender differences; however, all of them were associated with a trivial effect size (<1%), suggesting no practical significance for any of them.

As shown, profile membership was substantially discriminative for all ESEM factor scores, with partial η^2^ of associated to principal univariate effects ranging from 0.188 to 0.655 (see Table 4). With regards to univariate gender differences, non-significant effects were found for peer support, and for the other dimensions, women scored slightly lower than men on role clarity and management support and showed slightly higher scores on control, demands and poor relationships. Additionally, in this case, effect sizes associated with these differences were very low (all partial η^2^ lower than 0.01).

### 3.3. Study 2: Discussion

The results from Study 2 indicate some important ways in which men and women may differ in their exposure to psychosocial stressors but also a high degree of equivalence. Using a large sample of Italian workers employed in the public administration (PA) sector, LPA revealed that both men and women were optimally represented by three psychosocial risk profiles (i.e., normative, virtuous and at-risk), reflecting medium, low and high risk of exposure to stressors, respectively. The shape of the three profiles was similar for men and women, suggesting that exposure to positive and negative psychosocial factors was experienced in the same way across genders in different psychosocial risk profiles. This result suggests the absence of specific gendered configurations and risks concerning exposure to the psychosocial work environment, and it is consistent with Study 1 findings since no specific gendered configuration emerged from our LPA analyses. In this sense, we concluded that Study 2 revealed three profiles that could be qualitatively generalised across genders. Moreover, as documented by post-hoc analyses, results suggest that gender differences in stress risk exposure may not emerge in sectors with reduced gender segregation, such as PA.

However, the results from cross-tabulations between gender and profile membership showed that men had a higher probability than women of being classified in the *virtuous* profile, and the opposite pattern was detected for the *average* profile, while men and women displayed the same likelihood of being classified in the *at-risk* profile. This latter evidence suggested that, when exposed to overarching ‘extreme’ psychosocial environments, men and women appraised them equally (at least, in the PA sector) since no gender unbalance emerged for this profile. Consistent with gender role theory, this finding suggests that gendered resources (e.g., biological, personal, social and contextual) may not provide differential protective functions when employees are exposed to highly stressful psychosocial environments [6].

With regards to the virtuous profile, the larger proportion of men documented by Study 2 results may be in line with gender role theory, with men gaining more satisfaction (and less stress) than women in paid work [29] when exposed to a positive psychosocial environment. Moreover, a larger proportion of men clustered in the virtuous profile may signal that women appraise positive psychosocial environments less frequently than men, since they are more cogently exposed to multiple social roles (e.g., family and caregiving responsibilities [45], and the accumulation of these demands may attenuate their overall experience of positive work environments compared to men [46]. However, the discrepancy between the two gender proportions in the virtuous profile might not be high, as one would expect from a more general perspective. This may be because the public administration sector is more gender balanced than others, and such differences may be attenuated [34]. The average profile reported a higher proportion of women than men, in contrast to the gender pattern observed for the virtuous profile. This corresponding pattern can be seen as the other side of the coin, with men being more likely to derive more enjoyment and less stress from work and therefore in the virtuous profile.

Finally, the analysis of simple effects revealed that women perceived a less favourable work environment than men, both on *average* (i.e., higher job demands and lower management support) and in the *virtuous* profile (i.e., higher job demands, lower control, role clarity and management support), but they scored higher than men in role clarity and control in the *at-risk* profile (and higher scores also in poor relationships). However, we highlighted that the size of the effects associated with simple effects was systematically trivial (in all cases, partial η^2^ < 0.01), and their statistical significance was mainly due to the very high sample size implied in the study. In this sense, we concluded that the interaction between configuration and gender groups in shaping specific exposures to psychosocial environments was substantially absent.

## 4. General Discussion and Conclusions

Overall, both Study 1 and Study 2 supported previous findings that gender differences in exposure and experience of stress are inconsistent. Study 2 found that in a working context where participation is gender balanced and roles are not expected to be divided along gender lines, both men’s and women’s experiences can be characterised by three similar profiles. Where we found a significant difference is in the propensity for men to experience the work environment as ‘virtuous’, whereas women’s experience is more likely to be in the ‘normative’ profile. The lack of gender differences in the probability of being ‘at risk’ of exposure to work stressors underlines the importance of addressing low-quality work in general rather than considering help for at-risk workers along gendered lines. The literature on gender role theory and the gendering of work and organisations more generally suggests that men and women are likely to be exposed to different stressors and to experience different stress profiles, but we found little support for this.

It should be acknowledged that in Study 1, we reviewed literature examining gender differences in important workplace stressors identified by the literature rather than adopting a more open-ended approach and we excluded studies looking at work–family balance. However, a recent meta-analysis examining work–family balance [20] demonstrated that gender differences in exposure cannot be assumed and it is difficult to make generalisations. Comparing the results of Study 2 with Study 1 suggests that working context does matter and in occupational settings with less gender segregation, gender differences in psychosocial working environment are limited. The data included in our research were collected before the COVID-19 emergency and did not take into account the associated changes in the way the work is organised and potential emergent stressors. Recent European data have highlighted some emergent gender differences in some stressors during the COVID-19 pandemic, as women with demanding jobs feel higher work/family conflict due to increased working hours from home [47]. Thus, future studies must also consider potential factors related to emergent trends in work that might influence different gender profiles. It might be useful to include further variables that can account for gender differences and to stressors related to age, marital status, education, having children, family role and caring responsibilities. It is important to adopt an intersectional approach and consider gender together with other variables (such as ethnicity and religion), which might further the understanding of gender’s role in relation to workplace stress and wellbeing.

While there were some limitations to Study 2 in that it was not longitudinal and we were not able to assess stress outcomes alongside exposure, it also had some methodological strengths. The large sample included several nested organisations and was gender balanced and representative of the Italian population as a whole. In Study 2, we did not analyse worker perceptions in a workplace context that was highly gendered, and while the review carried out in Study 1 did incorporate studies looking at gender-dominated workplaces, the findings overall did not support strong gender differences in exposure to stressors. A clear next step would be to compare a working context like that explored in Study 2 with gender-dominated contexts for men and women, respectively and to further explore the impact of gender differences at the role and occupational levels within these contexts, which we were unable to do in this study. In addition, future studies should investigate the moderating role of gender in the association between stressors and wellbeing outcomes to investigate if the impact of certain stressors on certain outcomes could be gender specific.

The findings also highlight interesting aspects for practical implications. In gender-balanced contexts, psychosocial risk management in the workplace should focus primarily on organisational and work aspects rather than on gender-specific interventions. In such contexts as public administration, an effective assessment and management of risks associated with work-related stress must primarily consider different tasks, jobs or organisational roles. Even though we found relatively few gender differences, men were significantly more likely to experience work as ‘virtuous’, which is in line with the literature asserting that workplace norms are gendered in favour of men. Practically addressing this gender difference is not straightforward since such gendering of the workplace is likely to be entrenched, but the lack of further differences does suggest that perhaps norms have and are shifting to become more gender balanced. Shaping the workplace such that women are as likely to find it as rewarding as men is also only one side of the equation since literature on work–life balance highlights how the demands of life outside of work may shape work stress. In this regard, it should be specified that in Italy in the public administration sector, a series of welfare initiatives have been introduced at the company level in recent years to specifically improve working conditions (e.g., maternity protection, company crèches, childcare leaves, and flexible work arrangements).

## Figures and Tables

**Figure 1 ijerph-20-05541-f001:**
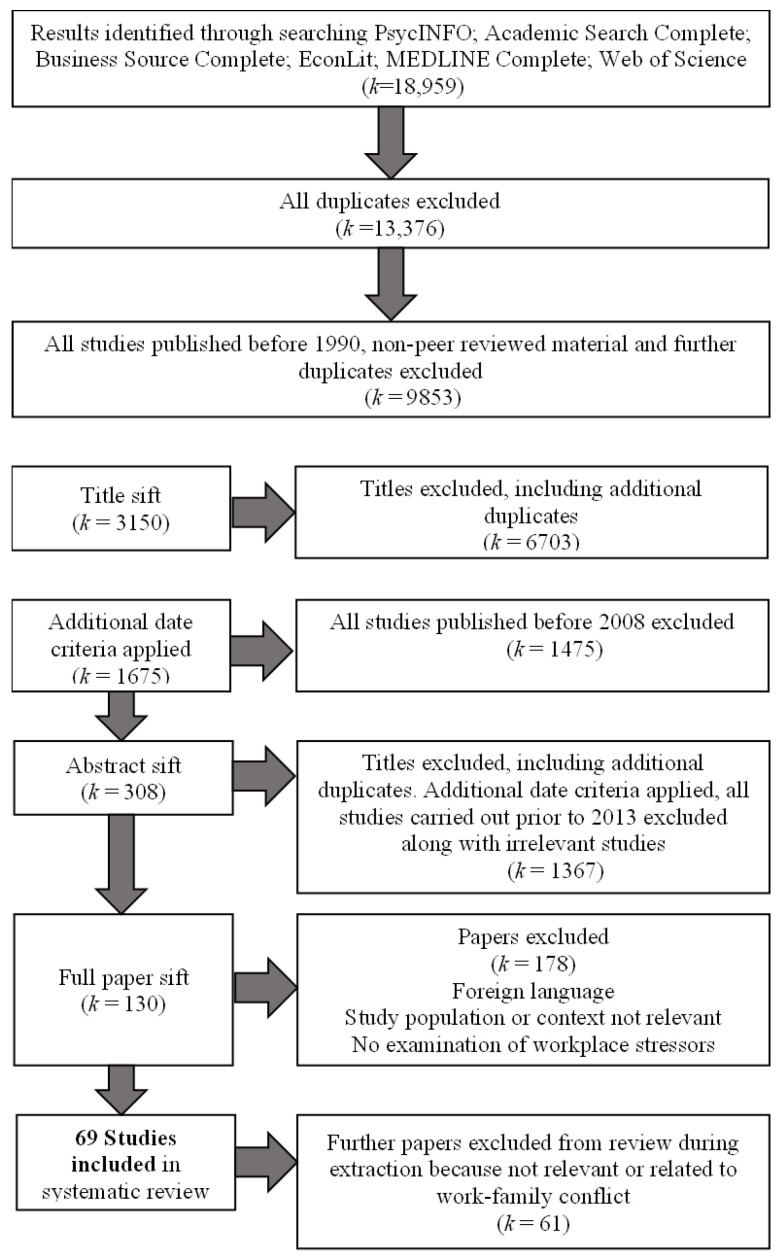
Flowchart of Search Process.

**Figure 2 ijerph-20-05541-f002:**
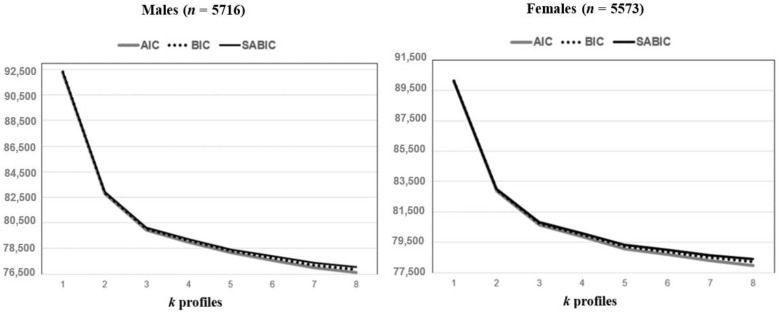
Elbow Plot of the Information Criteria Values.

**Figure 3 ijerph-20-05541-f003:**
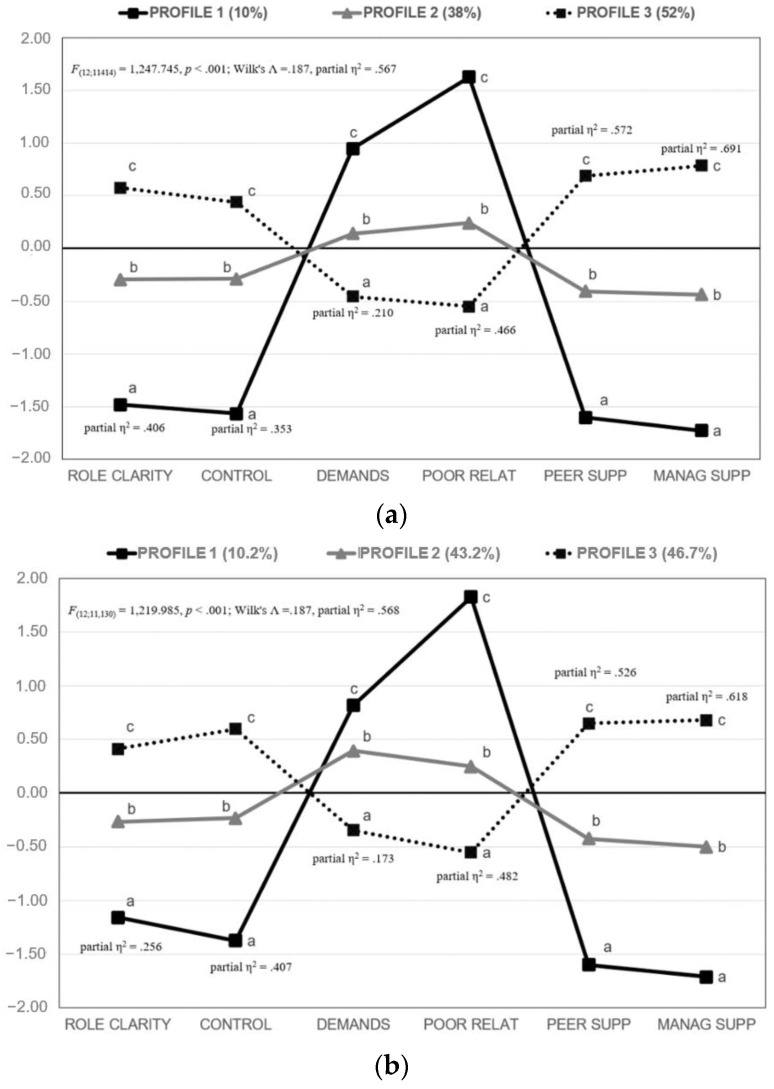
Final 3-Profile Solution for Men and Women. Note. Plotted scores were completely standardised for the full sample. Different subscripts reflect significant differences in post-hoc tests (Tukey method). All principal effects were significant for *p* < 0.001. (**a**) Men (*n* = 5716) (**b**) Women (*n* = 5573).

**Table 1 ijerph-20-05541-t001:** Descriptive statistics among the ESEM-MS-IT dimensions.

Means, SDs, Factor Correlations and Reliability Coefficients (Men)								
	M	SD	1.	2.	3.	4.	5.	6.
1. Role Clarity	0.001	0.829	0.84/0.80					
2. Control	−0.054	0.871	0.43	0.80/0.82				
3. Demands	−0.053	0.876	−0.22	−0.26	0.84/0.84			
4. Poor Relationships	−0.006	0.808	−0.30	−0.39	0.37	0.70/0.75		
5. Peer Support	−0.006	0.750	0.46	0.42	−0.35	−0.44	0.84/0.83	
6. Management Support	0.003	0.727	0.51	0.43	−0.32	−0.52	0.65	0.83/0.89
**Means, SDs, Factor Correlations and Reliability Coefficients (Women)**								
	**M**	**SD**	**1.**	**2.**	**3.**	**4.**	**5.**	**6.**
1. Role Clarity	−0.001	0.847	0.81/0.79					
2. Control	0.055	0.827	0.35	0.79/0.80				
3. Demands	0.055	0.925	−0.15	−0.31	0.84/0.85			
4. Poor Relationships	0.007	0.845	−0.19	−0.35	0.22	0.73/0.77		
5. Peer Support	0.006	0.834	0.29	0.41	−0.26	−0.46	0.83/0.83	
6. Management Support	−0.003	0.774	0.42	0.44	−0.28	−0.48	0.55	0.86/0.89

Note. M and SDs refer to the ESEM factor scores. All factor correlations are significant for *p* < 0.001. McDonald’s ω and Maximal Reliability *H* coefficients are reported along the diagonal of each factor correlation matrix (*H* coefficients are italicised).

**Table 2 ijerph-20-05541-t002:** Fit Indices of the LPA Solution for Each Gender Group.

k	*#Par*	Men (*n* = 5716)					Women (*n* = 5573)				
		LL	AIC	BIC	SABIC	Entropy	LL	AIC	BIC	SABIC	Entropy
1	12	−46,102	92,228	92,308	92,308	-	−45,023	90,069	90,149	90,111	-
2	19	−41,372	82,782	82,909	82,848	0.870	−41,420	82,878	83,004	82,944	0.815
3	26	−39,952	79,956	80,129	80,047	0.824	−40,299	80,650	80,822	80,739	0.785
4	33	−39,471	79,007	79,227	79,122	0.766	−39,908	79,883	80,101	79,996	0.790
5	40	−39,049	78,178	78,444	78,317	0.815	−39,499	79,078	79,343	79,216	0.766
6	47	−38,740	77,575	77,887	77,738	0.799	−39,299	78,693	79,004	78,855	0.780
7	54	−38,458	77,023	77,382	77,211	0.790	−39,096	78,301	78,658	78,487	0.791
8	61	−38,260	76,641	77,047	76,853	0.7900	−38,933	77,989	78,393	78,199	0.792

Note. *k* = number of profiles tested in the solution; *#Par* = number of estimated (free) parameters; LL = log-likelihood; AIC = Akaike Information Criterion; BIC = Bayesian Information Criterion; SABIC = Sample Size-Adjusted BIC.

**Table 3 ijerph-20-05541-t003:** Gender Differences in Profile Membership Distribution.

	Men		Women	
	%	Standardised Residuals	%	Standardised Residuals
Profile 1 (At-Risk)	5.1	−0.2	5	0.2
Profile 2 (Average)	19.2	−3.1	21.3	3.1
Profile 3 (Virtuous)	26.4	2.9	23	−2.9

Note. Percentages refer to the total sample (*n* = 11,289).

**Table 4 ijerph-20-05541-t004:** Principal Effects of the Two-Way Factorial MANOVA.

Factor	Variable	df_1_	df_1_ (df_2_ = 11,283)	F	*p*	Partial η^2^
Profile Membership	Role Clarity	2	11,283	2778.470	<0.001	0.330
	Control	2	11,283	3424.436	<0.001	0.378
	Demands	2	11,283	1304.807	<0.001	0.188
	Poor Relationships	2	11,283	5088.531	<0.001	0.474
	Peer Support	2	11,283	6852.763	<0.001	0.548
	Management Support	2	11,283	10,688.022	<0.001	0.655
Gender	Role Clarity	1	11,283	9.906	0.002	0.001
	Control	1	11,283	51.896	<0.001	0.005
	Demands	1	11,283	13.591	<0.001	0.001
	Poor Relationships	1	11,283	15.561	<0.001	0.001
	Peer Support	1	11,283	1.228	0.268	0
	Management Support	1	11,283	13.021	<0.001	0.001
Profile Membership X Gender(Interaction)	Role Clarity	2	11,283	48.137	0.000	0.008
	Control	2	11,283	6.824	0.001	0.001
	Demands	2	11,283	22.671	0.000	0.004
	Poor Relationships	2	11,283	9.678	0.000	0.002
	Peer Support	2	11,283	0.646	0.524	0
	Management Support	2	11,283	5.608	0.004	0.001

## Data Availability

The data that support the findings of this study are available from INAIL, but restrictions apply to the availability of these data, which were used under licence for the current study and are not publicly available. Data are however available from the authors upon reasonable request and with permission of INAIL.

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
