# Peer review of "Is Gender an Antecedent to Workplace Stressors? A Systematic Review and an Empirical Study Using a Person-Centred Approach"

_ijerph, 2023, doi:10.3390/ijerph20085541_

Round 1

Reviewer 1 Report

This paper presents interesting and valuable research. I’m familiar with many studies about gender, which tend to focus entirely on women at the expense of men. I consider your combined objectivity and comprehensiveness a major strength.

My major concern is that the manuscript has far exceeded the word limit. Reading like a short thesis than an academic paper, I stopped reading on Page 10 and scanned the rest. Please do not be discouraged. May I suggest considering publishing the two studies as separate manuscripts to be reconsidered for submission? As such, splitting this writing to present one article as a systematic review and the other as empirical research.

Occasionally, some sentences require shortening. My notes below may be helpful. Presenting two or three ideas in a long sentence is difficult reading. One idea per sentence of 20-30 words is ideal. As such, minor edits will assist reading fluency.  

Some minor notes:

- According to the APA format, numerals in the text under ten are spelt (e.g., ‘four’ instead of '4') unless statistics are used.

-                   - Starting a sentence with a numeral also needs spelling out (e.g., Seventeen instead of 17 – see page 8).

-                      - A study in the text does not have a comma when multiple authors are cited (e.g., “Gascoigne et al., (2015)” should be “Gascoigne et al. (2015)”.

Suggested edits: 

Abstract

Page 1 - Define ‘stresses’, ‘virtuous’, and ‘average’

 Page 2 – “Whilst findings are inconsistent, … of stressors to vary”.

Suggested edit:

“Whilst findings are inconsistent, different reactions indicate gender influences on how stress is perceived. Consequently, we expect perceptions of stressors to vary”.

 Page 2 – Break the paragraph here ‘profession. Gender’

Pages 2 and 3 – “Whilst both hypotheses … inferred (Jarman et al., 2012) – This sentence is too long. Delete ‘and’ insert a period, and write “Despite clear …”

 Page 3 Delete “In shaping exposure and vulnerability to workplace stressors”. Start with “Gender’s influence …”   

 Page 3 – “Accordingly, this study set out to inves-tigate whether the exposure to psychosocial risk factors differs for men and women?”  Remove question mark. Or state “this study sought to answer the question: …”

 Page 3 Delete “We examine this question through a systematic literature review and quantitative analysis of Italian public sector workers”. It’s stated below.

 Page 8 – “For all stressors a significant proportion of studies find no significant gender differences and the review found studies presenting evidence of greater exposure for both men and women in relation to all stressors”.

 Suggested edit - “The review found no significant gender differences for all stressors”.

 Page 8 – “Furthermore, it could also be argued that the gendered nature of work may be more general, in which case more detailed examination of a sector which is relatively gender balanced in composition and gender neutral in the experience of work supports a rigorous testing of this proposition”.

Unclear

 Page 8 In her seminal article, Acker (1990), argues that work and organisations are in fact gendered in a more general sense, that the organisation of work is premised on the division of men as breadwinners and paid workers and women as unpaid caregivers (see also Gascoigne et al., 2015).

 Page 9 – “PA worksites” Unclear. Spell out.

 Page 10 - (Istituto Nazionale di Statistica (ISTAT), 2017).

Author Response

See the attached document

Reviewer 2 Report

The introduction is too long. Please make it as 1 page.

Author Response

See the attached document

Reviewer 3 Report

I have read with interest this paper discussing the gender differences in exposure to stressor. Authors found that there are only little empirical supports to the theory of gendering of work and organizations.

I want to compliment the authors on the comprehensiveness of the literature review and on the originality of the issue.

The work is well written and results are clearly exposed in the section.

At page 8, in the section "Study 1: Discussion" line 4 replace "find" with "found".

At page 9, in the section "Study 2", third paragraph, remove "analysis".

Authors argued that there is a lack of consistent evidence in how gender might influence exposure to key workplace stressors, however in our experience some working factors, that are directely related to both gender and occupation should influence the physiological and pathological response to job stress. For example night shift work is a well recognized factor influencing the human response to stressors. Authors should better discuss, in my opinion, the possible negative health effects of night-work related stress (doi:10.23749/mdl.v111i3.9197  doi:10.3389/fpubh.2022.849310  doi:10.1016/j.numecd.2020.05.028) and also consider the different representation of gender among night workers (doi: 10.5271/sjweh.3520)

Author Response

see the attached document
